# Statistical Features in High-Frequency Bands of Interictal iEEG Work Efficiently in Identifying the Seizure Onset Zone in Patients with Focal Epilepsy

**DOI:** 10.3390/e22121415

**Published:** 2020-12-15

**Authors:** Most. Sheuli Akter, Md. Rabiul Islam, Toshihisa Tanaka, Yasushi Iimura, Takumi Mitsuhashi, Hidenori Sugano, Duo Wang, Md. Khademul Islam Molla

**Affiliations:** 1Department of Electronic and Information Engineering, Tokyo University of Agriculture and Technology, Tokyo 184-8588, Japan; sheuli@sip.tuat.ac.jp; 2Institute of Global Innovation Research, Tokyo University of Agriculture and Technology, Tokyo 184-8588, Japan; rabiul@sip.tuat.ac.jp; 3Department of Electrical and Electronic Engineering, Tokyo University of Agriculture and Technology, Tokyo 184-8588, Japan; wangduo@sip.tuat.ac.jp; 4Department of Neurosurgery, Epilepsy Center, Juntendo University, Tokyo 113-8421, Japan; yiimura@juntendo.ac.jp (Y.I.); tmituha@juntendo.ac.jp (T.M.); debo@juntendo.ac.jp (H.S.); 5RIKEN Center for Advanced Intelligence Project, Tokyo 103-0027, Japan; 6RIKEN Center for Brain Science, Saitama 351-0106, Japan; 7School of Computer Science and Technology, Hangzhou Dianzi University, Hangzhou 310018, China; 8Department of Computer Science and Engineering, Rajshahi University, Rajshahi 6205, Bangladesh; khademul.cse@ru.ac.bd

**Keywords:** epilepsy, seizure, seizure onset zone (SOZ), filter bank, feature extraction

## Abstract

The design of a computer-aided system for identifying the seizure onset zone (SOZ) from interictal and ictal electroencephalograms (EEGs) is desired by epileptologists. This study aims to introduce the statistical features of high-frequency components (HFCs) in interictal intracranial electroencephalograms (iEEGs) to identify the possible seizure onset zone (SOZ) channels. It is known that the activity of HFCs in interictal iEEGs, including ripple and fast ripple bands, is associated with epileptic seizures. This paper proposes to decompose multi-channel interictal iEEG signals into a number of subbands. For every 20 s segment, twelve features are computed from each subband. A mutual information (MI)-based method with grid search was applied to select the most prominent bands and features. A gradient-boosting decision tree-based algorithm called LightGBM was used to score each segment of the channels and these were averaged together to achieve a final score for each channel. The possible SOZ channels were localized based on the higher value channels. The experimental results with eleven epilepsy patients were tested to observe the efficiency of the proposed design compared to the state-of-the-art methods.

## 1. Introduction

Epilepsy is one of the most common chronic diseases of the brain [1,2]. According to the World Health Organization (WHO), it is the second most common neurological disorder and fifty million people worldwide have been diagnosed with epilepsy [3,4]. It is defined as sudden and repeated seizures, which are distinguished as either focal or generalized based on the propagation of brain activity [3,4,5]. To control seizures when anti-epileptic drugs fail, surgical treatment is a possible solution to prevent seizures by removing the area of the cortex from where the seizures are initiated, also known as the seizure onset sone (SOZ) [6]. For surgical treatment, SOZ localization from a part of the irritable and symptomatic zone is a difficult task and is time-consuming for epileptologists when done by visual examination of the multi-channel long-term intracranial electroencephalogram (iEEG) signals. Moreover, there is a lack of clinical experts for such a diagnosis; therefore, there is a high demand for the detection of SOZ channels via computer-aided systems in the respective community.

In different epilepsy studies, the use of feature extraction methods to design a machine learning system was a measure of statistical dispersion from continuous EEG signals. Gotman et al. [7] proposed the first widely applicable method of seizure detection by simple calculation of amplitude from EEG signals. Khan et al. proposed a wavelet-based seizure detection method from intracerebral EEG [8]. Their proposed feature extraction methods included energy, relative amplitude (RA) and coefficient of variation (CV), to capture the rhythmic nature of seizure discharges. Recently, Li et al. proposed CV and fluctuation index (FI) as features with a support vector machine (SVM) method to detect the seizure patterns in EEG [9]. Hassan et al. used the ellipse area of a second-order difference plot (SODP), CV, and FI as a feature extraction method to improve the system’s performance [10]. The studies listed above collectively suggested the use of statistical features; however, their studies only focused on detecting an EEG segment with either a seizure or non-seizure pattern.

A frequency domain or frequency subband is an important factor in the analysis of epileptic EEG. To design a computer-aided solution to detect epileptic focus, the existing studies can be categorized into low-frequency components (LFCs) [11,12,13,14,15] and high-frequency components (HFCs) [16,17]. In the LFC studies, Sharma et al. introduced an information theoretic feature called sample entropy, which is the input to the least square support vector machine (LS-SVM) classifier to discriminate focal and non-focal features [11]. Furthermore, the study was extended by the same authors [12] using different types of entropy features including Shannon, Renyi, Tsallis, Fuzzy, permutation, and phase entropies. Most recently, variants of decomposition methods such as wavelet, empirical mode decomposition (EMD), and bivariate-EMD (BEMD) have been applied with several statistical and entropy measures to extract features from iEEG signals, improving the performance of the system [14,18,19,20,21]. For instance, Acharya et al. suggested 23 feature extraction methods, including statistical and entropy features to design automated systems [22]. A least-squares support vector machine (LS-SVM) classifier was used to detect the focal signals. The above LFC studies commonly used the public Bern–Barcelona dataset, which consists of a pair of focal and non-focal iEEG signals with a lower frequency bandwidth (0.5 to 150 Hz).

Recent studies have reported that HFCs consisting of ripple (80–250 Hz) and fast ripple (250–600 Hz) bands bear the potential to guide the surgical treatment of drug-resistant focal epilepsy [23,24,25]. In particular, the repetitive waveform patterns, called the high-frequency oscillations (HFOs), in the ripple (80–250 Hz) and fast ripple (250–600 Hz) bands are valid biomarkers to localize the SOZ for epilepsy surgery [26,27,28]. These findings provide evidence of the use of ripple and fast ripple bands in iEEGs, which limits the LFC-related studies. By considering this finding, several works have proposed HFO detectors to identify SOZ use thresholding approaches, such as root mean square (RMS), short-time line length, complex wavelet transforms, and Hilbert envelope [29,30,31,32,33] by identifying the number of HFOs in iEEG data. The most recent studies in HFOs by Zuo et al. [25] proposed a convolutional neural network (CNN)-based method for determining the two kinds of HFOs in ripple and fast ripple and compared them with four traditional methods used in the RIPPLELAB toolbox [34]. HFOs detected in such a way can be used as features to identify the SOZ channels. Varatharajah et al. [35] proposed the use of the ratio of HFOs as well as other biomarkers found with unsupervised learning as a feature for SVM in order to identify SOZ. However, SOZ identification based on biomarkers requires two-step machine learning detection—identifying biomarkers and identifying SOZ. Moreover, the above studies focused on detecting HFOs in ripple and fast ripple iEEG data separately and therefore require long-term iEEG data to calculate the baseline.

Apart from the use of such a biomarker, our recent study [16] suggested that information theoretic entropies in HFCs are still useful in identifying SOZ channels. Reference [16] employed eight types of entropy-based feature extraction methods for SOZ identification. However, the significant shortcomings of using entropy features to design computer-aided solutions are that, first, the values of entropy measures strongly depend on the choice of parameter selection [36,37]. For entropy measures, various entropy types use three or more parameters, and the choice of these parameters also depends on test data and their length [36,37]. Therefore, there are many possible combinations of parameter settings based on the data that enable us to design a method. Second, to utilize the HFC with a range from 100 to 600 Hz in each segment of multi-channel iEEGs, the number of sample points in each segment was large. This large number of sample points leads to a higher computational cost [16]. Moreover, the detection of SOZ channels in Reference [16] was achieved based on the number of detected segments on SOZ and non-SOZ channels, and the detected decision was performed using an SVM classifier based on a hard threshold.

To address the above problems, we hypothesized that simple statistical features are still efficient in SOZ localization because these statistical features have already been used in other epilepsy study contexts [7,8,9,10,17]. Thus, our contributions in this study as follows:We propose 12 feature extraction methods, consisting of nine statistical and three entropy measure features that are important to characterize epileptic signals. Several studies of epilepsy used these statistical features to characterize normal and epileptic brain activities [7,8,9,10], which provide evidence to support the use of statistical features. To the best of our knowledge, this is the first time an investigation has been carried out to detect the SOZ electrodes based on statistical methods using HFCs (ripple and fast ripple bands).A data-driven grid search method using MI scores is developed to optimize bands and features jointly. The joint selection of appropriate bands and features related to epileptic activities may improve the performance of the proposed computer-aided solution and this joint selection has not been reported in previous SOZ detection studies.We compared different designs with SVM and standard state-of-the-art LightGBM classifiers and selected an optimal method to provide a graphical representation for epileptologists to identify the possible SOZ channels. The SOZ detection was performed based on the scoring for each segment of channels estimated using the LightGBM algorithm.

To evaluate the method, we analyzed the iEEGs of 11 patients by way of time series prediction. The proposed method’s methodological framework will be more practical for use in clinical applications to identify possible seizure onset channels. To date, there has been no formal comparison between patient-dependent and -independent design in HFCs. To assess the possibility of building a patient-independent design (PID), we also compared the proposed framework’s performance with patient-independent design.

## 2. Data and Methods

### 2.1. Dataset

The dataset used to evaluate this study’s performance is collected from the Epilepsy Center of Juntendo University, Tokyo, Japan. The study was jointly approved by the Ethics Committee of Juntendo University Hospital and the Ethics Committee of the Tokyo University of Agriculture and Technology. To record data, the epilepsy surgeon implanted platinum subdural grids (UNIQUE MEDICAL Co, Tokyo, Japan) with 4-mm diameters and 10-mm distances for the cortical surface and platinum strip electrodes (UNIQUE MEDICAL Co, Tokyo, Japan) with 3-mm diameters and 5-mm distances for the vertical direction and the bottom of the cortex. The dataset consists of long-term iEEG collected from adult and pediatric patients with focal epileptic seizures. The data were collected from eleven patients with medically intractable epilepsy caused by focal cortical dysplasia (FCD) represented as Pt1, Pt2, Pt3, Pt4, Pt5, Pt6, Pt7, Pt8, Pt9, Pt10, and Pt11. The dataset includes males and females between 5 and 39 years. Table 1 shows the summary of the dataset, including patient ID, age and sex, lesion site, pathology, location, sampling frequency, number of electrodes, number of seizure onset zone (SOZ) electrodes, follow up, and Engel epilepsy surgery outcome scale [38]. This study used the interictal iEEG data of 1 h (one hour) length with sampling frequencies of 2 kHz and 1 kHz for eight and three patients, respectively (see Table 1). In the case of the bottom of sulcus (BOS) patients, the epileptologist dissected the cortical sulcus and put in small electrodes on the vertical sulcus. The iEEG dataset was recorded using the Neuro Fax digital video EEG system (NIHON-KODEN, Tokyo, Japan) with a length of several days. The epileptologists selected the sleep stage iEEG without motion artifacts for analysis. They assigned the positive labels to the SOZ electrodes for each patient and a negative label was given to the rest of the electrodes. Therefore, data obtained from SOZ and non-SOZ electrodes were used to design the proposed computer-aided solution. The seizure-free outcomes were treated as class IA in Engel’s classification. The mean follow up period was 4.9 ± 1.0 years. Seizure outcomes were evaluated using Engel’s classification at the last visit to the outpatient center. All the patients signed an informed consent paper before the recording of iEEG data. As an example, Figure 1 shows the row interictal iEEG data with 17 electrodes and their corresponding MRI images for patients Pt1 and Pt6. The electrodes with a red color represent the SOZ channel.

### 2.2. Segmentations and Filter Bank Analysis

In this study, one-hour multi-channel interictal iEEG was split into 20 s segments. Each segment was filtered with a third-order Butterworth bandpass filter to extract the high-frequency components from interictal iEEG.

Several studies have proposed subband decompositions that enhance the detection results. In those studies, multiple filters with different passbands were used [17,40,41]. Epileptic focus detection with a filter bank approach was first proposed by our recent study, where the HFCs (ripple and fast ripple) were divided into multiple subbands [16,17]. The present study also used a similar subband technique to divide the HFCs (ripple and fast ripple) into multiple subbands, each of which has a bandwidth of 50 Hz. The bandpass filters for extracting subband components $S_{n}$, where n=1,2,…,N, from iEEG signals were third-order Butterworth bandpass filters. In our case, the total number of subbands *N* was 10 for eight patients with sample frequency of 2 kHz and 7 for three patients with a sample frequency of 1 kHz. In other words, the cut-off frequencies were from 100–450 Hz for patients with 1 kHz sample frequency and the cut-off frequencies for patients with 2 kHz were from 100–600 Hz, which cover ripple and fast ripple bands in HFOs.

### 2.3. Feature Extraction Methods

Feature extraction is an important step in designing a computer-aided system. In different bio-signal processing research [7,8,9,10,17,42,43], such as iEEG, EMG, and EEG, a combination of the following feature extraction methods were proposed. To extract the features from each segment of iEEG signals, let us define each channel of the *n*-th subband as x, which can be represented as $x=\left[x_{1},x_{2},\ldots ,x_{L}\right]$, where *L* is the length of x. We introduce several feature extraction methods, which are described in the following subsections.

#### 2.3.1. Coefficient of Variation

The coefficient of variation (CV) was used to analyze the characteristics of iEEG signals in different epilepsy studies [8,9,10,17] that measure the dispersion of data. The CV indicates the ratio of standard deviation to the mean, and it provides information about variations in any signal amplitude. For x, the coefficient of variation is defined as [9,10]:(1)$$\begin{array}{c} CV=\frac{\sigma }{\mu }, \end{array}$$
where μ and σ represent the mean and standard deviation of x, computed as:(2)$$\begin{array}{c} \mu =\frac{1}{L}\sum_{i=1}^{L}\left|x_{i}\right|, \end{array}$$
(3)$$\begin{array}{c} \sigma =\sqrt{\frac{1}{L}\sum_{i=1}^{L}{\left(x_{i}-\mu \right)}^{2}}, \end{array}$$

#### 2.3.2. Fluctuation Index

The fluctuation index (FI) is used to measure the intensity of signal amplitude changes applied in different epilepsy studies for seizure detection [9,10,44]. FI is defined from x as:(4)$$\begin{array}{c} FI=\frac{1}{L-1}\sum_{i=1}^{L-1}\left|x_{i+1}-x_{i}\right|, \end{array}$$

#### 2.3.3. Variance

The variance (Var) of a signal refers to measuring how far the amplitudes of the signal are spread out from their average value, and is also used to characterize the nature of seizure and non-seizure signals with the Bonn dataset [17,45,46,47]. The variance of x can be calculated as:(5)$$\begin{array}{c} Var=\frac{1}{L-1}\sum_{i=1}^{L}{\left(x_{i}-\mu \right)}^{2}, \end{array}$$

#### 2.3.4. Root Mean Square

The root mean square (RMS) is one of the popular features used to characterize HFOs in ripple and fast ripple iEEG-related studies to identify SOZ [17,29,34,48]. In mathematics, it can be defined as:(6)$$\begin{array}{c} RMS=\sqrt{\frac{1}{L}\sum_{i=1}^{L}{x_{i}}^{2}}, \end{array}$$

#### 2.3.5. Difference Absolute Standard Deviation

The difference absolute standard deviation (DASD) is another popular statistical method to extract features used in biomedical signal processing studies [17,49,50]. It resembles the RMS feature and can be expressed as:(7)$$\begin{array}{c} DASD=\sqrt{\frac{1}{L-1}\sum_{i=1}^{L-1}{\left(x_{i+1}-x_{i}\right)}^{2}}, \end{array}$$

#### 2.3.6. Mean Absolute Value

The mean absolute value (MAV) of a signal is the average of the summation of absolute value, which is used in the characterization of bio-signals [51]. For x, MAV can be defined as:(8)$$\begin{array}{c} MAV=\frac{1}{L}\sum_{i=1}^{L}\left|x_{i}\right|, \end{array}$$

#### 2.3.7. Modified Mean Absolute Value

The modified mean absolute value (MMAV) is an extended version of the MAV feature. A window is defined with two discrete values and the signal is weighted by the window. The MMAV of x can be defined as [42,52,53]:(9)$$\begin{array}{c} MMAV=\frac{1}{L}\sum_{i=1}^{L}w_{i}\left|x_{i}\right|,w_{i}=\left\{\begin{array}{cc} 1, & \;\mathrm{if}\;0.25L\leq i\leq 0.75L\\ 0.5, & \;\mathrm{if}\;\mathrm{otherwise} \end{array}\right. \end{array}$$
where $w_{i}$ is the weighting window.

#### 2.3.8. Modified Mean Absolute Value 2

The modified mean absolute value 2 (MMAV2) is another extension of mean absolute value. A window function weights the signal before calculating the MAV [50]. The window function is different from MMAV. It is expressed using x as:(10)$$\begin{array}{c} MMAV2=\frac{1}{L}\sum_{i=1}^{L}w_{i}\left|x_{i}\right|,w_{i}=\left\{\begin{array}{cc} 1, & \mathrm{if}\;0.25L\leq i\leq 0.75L\\ 4i/L, & \mathrm{if}\;i<0.25L\\ 4(i-L)/L, & \mathrm{otherwise} \end{array}\right.. \end{array}$$
where $w_{i}$ is the weighting window function for the signal *x*.

#### 2.3.9. Log Detector

The log detector (LD) was also used to extract features in biomedical signal processing [54]. Based on the logarithm and log detector (LD) feature, the nonlinear detector can be characterized as $\log\left(\left|x_{i}\right|\right)$. For x, the LD is expressed as [54]:(11)$$\begin{array}{c} LD=\exp\left(\frac{1}{L}\sum_{i=1}^{L}\log\left|x_{i}\right|\right). \end{array}$$

#### 2.3.10. Permutation Entropy

The permutation entropy (PE) is a simple and computationally robust method used for the prediction of epileptics seizure by estimating the complexity of time series [55,56]. At each time *i* for a given time series x, each vector with the *m*-th subsequent values is defined as:(12)$$\begin{array}{c} i\mapsto \left(x_{i},x_{i+1},\ldots ,x_{i+\left(m-1\right)}\right), \end{array}$$
where *m* is the embedding dimension. An ordinal pattern associated with this vector defines permutation as $\pi =\left(k_{0}k_{1}\ldots k_{m-1}\right)$ of $\left(0,1,\ldots ,m-1\right)$, which satisfies $x_{i+k_{0}}\leq x_{i+k_{1}}\leq \cdots \leq x_{i+k_{m-1}}$. By considering a time lag τ, Equation (Equation 12) can be further extended as:(13)$$\begin{array}{c} i\mapsto \left(x_{i},x_{i+\tau },\ldots ,x_{i+\left(m-1\right)\tau }\right). \end{array}$$

For each time series, there is a probability distribution π, whose elements $\Pi_{j}$ (j=1,2,…,m!) are the frequencies associated with the *j* possible permutation patterns. The PE can be defined as:(14)$$\begin{array}{c} PE=-\sum_{j=1}^{m!}\Pi_{j}\log_{2}\Pi_{j}. \end{array}$$ In this study, the parameters *m* and τ were set to 3 and 1, respectively.

#### 2.3.11. Spectral Entropy

Spectral entropies are used to measure the complexity of a time series based on the power spectrum [57]. Several epilepsy seizure-related studies [57,58,59] have proposed the use of spectral entropy, such as Shannon (ShE) and Reny’s entropy (RE). To compute spectral entropy, the normalized power $p_{f}$ was estimated using the Fourier transform (FT) of x and defined as:(15)$$\begin{array}{c} p_{f}=\frac{P_{f}}{\sum P_{f}}, \end{array}$$
where $P_{f}$ is the power level of the frequency component. The spectral entropy, including Shannon (ShE) and Reny’s entropy (RE), can be defined [57] as:(16)$$\begin{array}{cc} ShE & =-\sum_{f}p_{f}\ln\left(p_{f}\right), \end{array}$$(17)$$\begin{array}{cc} RE\left(\gamma \right) & =\frac{1}{1-\gamma }\sum_{f}\ln p_{f}^{2}, \end{array}$$
where γ is the order of RE’s entropy (γ=2).

### 2.4. Feature Concatenation

For x, we calculated each feature using the above feature extraction method from each segment and concatenated them sequentially to represent a vector form. Therefore, the feature vector $v_{n}$ of the *n*-th subband for a channel can be defined as:(18)$$\begin{array}{c} v_{n}=\left[u_{n}^{\left(1\right)},u_{n}^{\left(2\right)},\ldots ,u_{n}^{\left(D\right)}\right]\in {I\;R}^{D}, \end{array}$$
where *D* indicates the number of total features (in our case D=12).

### 2.5. Subband and Feature Selection Method

To select the relevant features and subbands, mutual information (MI) was used to estimate the scores of features in the training set. Let us denote training features with the *n*-th subband $M_{n}\in {I\;R}^{H\times D}$, where H=ch×s, such that ch and *s* are the total number of channels and segments, respectively. The training features $M_{n}\in {I\;R}^{H\times D}$ were estimated for all channels with each segment and finally stacked all of the segments. The MI from the set of training features $M_{n}$ and class *C* for the *n*-th subband is defined sequentially [60] as:(19)$$\begin{array}{c} MI\left(M_{n};C\right)=\sum_{m_{n}^{\left(i\right)}\in M_{n}}\sum_{c\in C}p\left(m_{n}^{\left(i\right)},c\right)\log\left(\frac{p(m_{n}^{\left(i\right)},c)}{p\left(m_{n}^{\left(i\right)}\right)p\left(c\right)}\right), \end{array}$$
where $m_{n}^{\left(i\right)}$ is the *i*-th feature of the *n*-th subband. $p(m_{n}^{\left(i\right)},c)$ is the joint probability of $m_{n}^{\left(i\right)}$ and *c*. $p(m_{n}^{\left(i\right)})$ and p(c) are the marginal probability density functions of $m_{n}^{\left(i\right)}$ and *c*, respectively. We used the bin method [60] to estimate MI score between the features $m_{n}^{\left(i\right)}$ and label *c* from each feature of the training set $M_{n}$. The MI scores $s_{n}$ from the training features $M_{n}$ of the *n*-th subband can be defined as:(20)$$s_{n}=\left[s_{n}^{\left(1\right)},s_{n}^{\left(2\right)},\ldots ,s_{n}^{\left(D\right)}\right]\in {I\;R}^{D}.$$

#### 2.5.1. Subband Scoring

From Equation (Equation 20), we can calculate the MI scores of the *n*-th subband as:(21)$${\bar{s}}_{n}=\frac{1}{D}\sum_{j=1}^{D}s_{n}^{\left(j\right)}.$$ The set of mutual information, ${\bar{s}}_{n}$, for all subbands was rearranged in descending order, such that ${\bar{s}}_{\lambda (1)}\geq \cdots \geq {\bar{s}}_{\lambda (n)}\geq \cdots \geq {\bar{s}}_{\lambda (N)}$, where λ(n) is the sorted index of the *n*-th subbands. The set of sorted MI scores for all subbands can be defined as:(22)$$S_{sr}=\left[{\bar{s}}_{\lambda (1)},{\bar{s}}_{\lambda (2)},\ldots ,{\bar{s}}_{\lambda (N)}\right]\in {I\;R}^{N}.$$

#### 2.5.2. Feature Scoring

The scores of the *d*-th feature can be defined as:(23)$${\bar{s}}^{\left(d\right)}=\frac{1}{N}\sum_{i=1}^{N}s_{i}^{\left(d\right)}.$$ The set of average mutual information, ${\bar{s}}_{d}$ for all features, was rearranged in descending order, such that ${\bar{s}}^{I(1)}\geq \cdots \geq {\bar{s}}^{I(d)}\geq \cdots \geq {\bar{s}}^{I(D)}$, where I(d) is the sorted index of the *d* feature. The set of sorted average MI feature scores for *D* features can be defined as:(24)$$F_{sr}=\left[{\bar{s}}^{I(1)},{\bar{s}}^{I(2)},\ldots ,{\bar{s}}^{I(D)}\right]\in {I\;R}^{D}.$$

Finally, we applied a grid search method between the subbands and features with the higher scores ($S_{sr}$ and $F_{sr}$) to estimate $F_{1}$ scores. The maximum $F_{1}$ score was used to select the optimal number of features and subbands. This metric also used to evaluate the system performance in epileptic focus detection [16]. The feature $V^{*}$ with optimal subbands *J* and features *K* for a channel is defined as:(25)$$\begin{array}{c} V^{*}=\left[{\tilde{v}}_{\lambda (1)},\ldots ,{\tilde{v}}_{\lambda (n)},\ldots ,{\tilde{v}}_{\lambda (J)}\right]\in {I\;R}^{J\times K}, \end{array}$$
where ${\tilde{v}}_{\lambda (n)}=\left[u_{n}^{I(1)},u_{n}^{I(2)},\ldots ,u_{n}^{I(K)}\right]$ for the *n*-th subbands.

### 2.6. Classifiers

#### 2.6.1. Support Vector Machine

The SVM uses an optimal hyperplane to separate data from two classes [61]. The RBF kernel nonlinearly maps data into a higher dimensional space so that it can handle nonlinear relationships between dependent and independent variables. The parameter of SVM was set based on the training set. To obtain the optimal hyperplane, an SVM with a radial basis function kernel (RBF) was used in this study.

#### 2.6.2. LightGBM

LightGBM uses the gradient-boosting decision tree algorithm for classification used in machine learning for epilepsy seizure detection [62]. Gradient boosting sequentially creates new models from an ensemble of weak models with the idea that each new model can minimize the loss function. This loss function is measured by a gradient descent method. The detailed derivation and additional information are available in Reference [63].

### 2.7. Evaluation

#### 2.7.1. Division of iEEG Time Series for Training and Testing

To evaluate the developed patient-dependent computer-aided solution, the division of the time series iEEG is the critical step. By considering the nature of the time series, we used time series cross-validation techniques [64,65,66], one of the system’s appropriate solutions to evaluate the model. We divided the iEEG data into training, validation, and testing by way of time series forecasting, as illustrated in Figure 2. In this study, we used 60 min interictal iEEG data divided into 20 s segments consisting of a total of 180 segments. We used 30 min iEEG data (90 segments total) to train both SVM and LGBM models. The 5 min iEEG data (15 segments) was used to tune the system parameters, and the 5 min data (15 segments) was unused. The data from the remaining 20 min (60 segments) was used in testing for our proposed system. Due to the imbalance of SOZ and non-SOZ channels in iEEG data, the number of non-focal segments was much compared to the number of focal segments. Several studies have observed the problems of using machine learning methods for the imbalanced distribution in minority and majority classes [67,68,69].

Our previous study for epileptic focus detection provided evidence that the imbalanced learning problem can deteriorate the performance of the system [16]. To solve the imbalanced learning problem, the adaptive synthetic (ADASYN) approach [69] was used. Let us define the training features $\nu \;=\;\left\{v_{F}^{*(i_{in})},v_{N}^{*(j_{in})}\right\}$ after selecting subbands and features induced from Equation (Equation 23), where $v_{F}^{*(i_{in})}$ and $v_{N}^{*(j_{in})}$ denote the feature vector of the $i_{in}$-th sample of the focal segment and the feature vector of the $j_{in}$-th sample of the non-focal segment, respectively, and $I_{in}\ll J_{in}$ due to the imbalanced dataset. The balance training set $\tilde{\nu }$ was defined in Reference [69] from the training set ν as:(26)$$\begin{array}{c} \tilde{\nu }=\left\{v_{F}^{*(i_{in})},v_{N}^{*(j_{in})},v_{F}^{*({\tilde{i}}_{in})}\right\}, \end{array}$$
where $I_{in}+{\tilde{I}}_{in}=J_{in}$. In this study, the parameters to balance the training set were set in a way similar to our previous study [16]. The balanced training set $\tilde{\nu }$ was the input of the SVM and the LightGBM method for training the model.

#### 2.7.2. Segment-Wise Performance Measurement

In this study, the performance evaluation metrics include sensitivity (Sen), specificity (Spe), and F-score ($F_{1}$ score) to measure the performance of segment-wise detection in SOZ and non-SOZ channels. The underlying idea behind showing statistics of segment-wise detection was that we provided measures for comparison studies similar to HFO- and low-frequency related works [24,25] used in Bern–Barcelona or other datasets. These evaluation metrics were also used in recent HFC-related research [16]. The calculations were as follows:(27)$$\begin{array}{c} Sensitivity=\frac{TP}{TP+FN}\times 100\%, \end{array}$$
(28)$$\begin{array}{c} Specificity=\frac{TN}{TN+FP}\times 100\%, \end{array}$$
(29)$$\begin{array}{c} F_{1}\;score=\frac{TP}{TP+0.5(FP+FN)}, \end{array}$$
where true positive (TP) denotes the number of correctly detected focal segments in the SOZ channels; false negative (FN) refers to the number of incorrectly detected non-focal segments in the SOZ channels; true negative (TN) indicates the number of correctly detected non-focal segments in non-SOZ channels; false positive (FP) means the number of incorrectly detected focal segments in the non-SOZ channels. A post-hoc test with the Bonferroni procedure was used to assess the statistical significance of the methods with a significance level of α(=0.05).

#### 2.7.3. Channel-Wise Performance Measurement

In this study, the main target was to design a method to identify the possible electrodes related to SOZ. To identify the electrodes with SOZ and non-SOZ, a final decision will come after observing the scores of multiple segments. Therefore, the performance of each patient was observed by AUC–ROC [70] by computing the sensitivity (Sen) and false positive rate (FPR) of the channels with each threshold value. In our case, we estimated scores from each segment of the test set and averaged them together to achieve the final score of the channels. After achieving sensitivity and false positive rate of the channels, we estimated the AUC by using the trapezoid rule [70].

## 3. Results

In this study, we developed a method to detect the SOZ channel on a case-by-case basis. For each of these cases, we investigated the improvement of the system performance with statistical evidence. First, we discussed the selection of optimal features with the most significant subbands in Section 3.1. To assess the effectiveness of the system, we compared several designs based on different features and classifiers and selected the best one, as discussed in Section 3.2. Considering the optimal method, we have provided an intuition with statistical measurements in Section 3.2 and Section 3.3. To evaluate the possibility of building patient-independent design (PID), we compared the proposed method with PID on data from eleven patients. The computational cost of extracting features from iEEG signals is vitally important for medical decisions to be made in a timely manner, and is discussed in Section 3.5. The parameters for designing the system with different features and classifiers are summarized bellow:**Filter bank Feature Extraction Method (FbFM):** The multi-channel interictal iEEG signals were split into 20 s segments. The *N* bandpass filters were implemented using a third-order Butterworth filter to decompose each segment of the high-frequency components (ripple and fast ripple) in iEEG. The different types of statistical feature extraction methods were applied to each subband to extract features. An SVM and LightGBM with the ADASYN method were used to score each electrode for identifying possible SOZ channels.**FbFM with Subband and Feature Selection (FBFM/Sb/FS):** In this case, subbanding and feature extraction were applied in the same way as the above **(FbFM)** method. A data-driven grid search method using MI scores was proposed to select both epileptic-related bands and prominent features. The ADASYN approach, SVM and LightGBM were also used to score channels.

### 3.1. Selection of Optimal Features and Subbands

A recent EEG-based study provided evidence that the selection of appropriate operational bands can significantly improve the system performance compared to the use of a wide range of frequency bands [40]. The challenging task in finding the theoretically optimal parameters is to select the prominent features with jointly epileptic-related bands. To address this problem, this study developed a data-driven grid search method based on MI scores to select the prominent features and epileptic subbands with joint contributions. The proposed method computes MI scores for each of the features and subbands. The subbands and features are rearranged according to their maximum scores. A grid search method was applied to select optimal features *K* and subbands *J* by aggregating the top ranked MI scores of subbands and features.

Figure 3 shows the F-score corresponding to the combination of optimal features and subbands. This figure shows the relationship between the F-score and the number of incorporated subbands and features with higher MI scores ($S_{sr}$ and $F_{sr}$ induced from Equation (Equation 22) and (Equation 24)), which may lead to improving the performance of the system. The optimal subands *J* and features *K* were adopted based on the maximum value of the F-score in the proposed computer-aided solution.

### 3.2. Results for Detected Segments

In this study, we proposed FbFM and FbFM/Sb/FS with SVM and state-of-the-art LightGBM classifiers to identify SOZ channels with eleven epilepsy patients. The performance of different methods in term of Sen, Spe, and $F_{1}$ score is a widely used metric to evaluate the system for an imbalanced dataset. Table 2 shows the experimental results for individual segment detection. The sensitivity (Sen) represents the detection of correctly predicted focal segments from the SOZ channels (Pt1: 82.78%; Pt2: 99.83%; Pt3: 67.77%; Pt4: 81.11%; Pt5: 90.83%; Pt6: 90.24%; Pt7: 83.83.00%; P8: 99.69%; P9: 89.33%; P10: 76.83%; and P11: 80.33%). In contrast, the specificity (Spe) is an evaluation metric used to characterize the rate of correctly predicted non-focal segments from the non-SOZ channels (Pt1: 99.44%; Pt2: 98.92%; Pt3: 90.09%; Pt4: 93.48%; Pt5: 98.20%; Pt6: 99.77%; Pt7: 98.78%; P8: 99.69%; P9: 98.20%; P10: 96.33%; and P11: 99.67%). To test the statistical significance of the methods (FbFM and FbFM/bS/FS), post-hoc test with $F_{1}$ scores were performed to observe the significance of the methods’ performance. From the results of the post-hoc tests, the FbFM/bS/FS method with LightGBM significantly outperformed the FbFM method (FbFM/Sb/FS vs. FbFM with SVM: p<0.05; FbFM/bS/FS vs. FbFM with LightGBM: p<0.05; FbFM/Sb/FS with LightGBM vs. FbFM/Sb/FS with SVM: p<0.05). A similar scenario is observed for the other evaluation metrics. Consistently, the proposed FbFM/Sb/FS approach with LightGBM achieves the highest performance for all patients. Considering the overall results, FbFM/Sb/FS with LightGBM was used as an optimal method for further analysis.

### 3.3. Results for Localizing SOZ channels

The previous section confirmed that the joint contribution of bands and feature selection can significantly improve the performance of the method. The performance of the optimal method (FbFM/Sb/FS) with LightGBM was observed in terms of the AUC for identifying SOZ channels. Figure 4 provides an illustration to hypothesize the possible SOZ channels by visually observing the value of scores over multiple segments much more easily and more reliably. The vertical axis in the color map (left) represents the electrodes, and the horizontal axis indicates the segment index with score values estimated using the proposed method (FbFM/Sb/FS) with LightGBM. It is observed that the strength of the scores in the segments of each SOZ channel is much higher than the non-SOZ channels (sharp yellow line). However, a final decision about the channels of SOZ can only be reached after observing the combination of multiple segments in the same channel. Therefore, it is necessary to measure a statistic with channel scores. Each channel score was achieved by averaging the scores across all of the segments, indicated by a bar plot on the right side with each color map of the patients. Table 3 shows the AUC results for identifying SOZ channels for each patient.

Figure 5 and Figure 6 represent the visualization of electrodes with MRI scan images. The “X” (lime color) represents the SOZ channels, labeled using clinical expertise, and the circles with color indicate the scores of the channels estimated by the proposed method. The values of the estimated scores for each channel were plotted onto the cortical surface. It is observed from both Figures that our proposed method provides higher scores (red color with circle in Figure 5 and Figure 6) for the electrodes labeled by clinical experts and also suggests some active electrodes that are close to the SOZ and may have been linked to seizure events.

### 3.4. Comparison Between Patient-Dependent and -Independent Designs

In fact, the patient-independent design (PID) of the system is more preferable for real-world applications compared to patient-dependent design (PDD). Therefore, it is desirable to compare the proposed PDD with PID to assess the possibility of designing a patient-independent system. Thus, in this study, we compared the proposed PDD and PID with AUC for localizing SOZ channels using the optimal classifier LGBM. For the PID, the feature extraction and model were created using the pooled training data of all subjects except the given test subject. Then, the resulting model was tested using the given data of test subjects for testing the model. To balance the class features, ADASYN with its default settings [16,69] was also applied to highly imbalanced feature sets in the training stage. The results obtained are displayed in Table 3 for both PDD and PID. Notably, since the cut-off frequency of the patients with a 1 kHz sample frequency (see Table 1) was between 100 Hz to 450 Hz, we used the bands between 100 Hz to 450 Hz for patients with a 2 kHz sample frequency to design the automated PID. To build our PID method for patients with a 2 kHz sample frequency, we removed the patients from the pooled training data with 1 kHz sample frequencies.

### 3.5. Computational Cost Analysis

The mean computational time for each feature extraction method (12 methods) using 10 subbands was measured by Python on an iMac Pro (with Intel Xeon W processor and 128 GB RAM). Notably, to detect a single segment at the testing phase, the mean results were estimated with 100 runs. Figure 7 shows a bar diagram (left) of the average computational time (in seconds) with each feature extraction methods for 10 subbands. From this figure, it is observed that the computational time of the MAV method is close to zero. Other methods such as FI, RMS, and VAR, the computational time is below 0.02 s. So, for a single-segment test, the average computational time with the combination of twelve feature extraction methods and 10 subbands was 0.75 s. Figure 7 also shows the average computational time (in seconds) with increased number of subbands for 12 feature extraction methods (right). It is observed that the computational time increases linearly as the number of subbands increases. Since the total computational time for a single segment test is 0.75 s, it is very convenient for clinical application and epileptologists can make quick medical decisions using long-term iEEG data.

## 4. Discussion

In this study, we used HFCs (>80 Hz) that include ripple and fast ripple bands for identifying SOZ channels. In a conventional clinical system, epileptologists visually inspect long-term data (typically more than three days, depending on the patient’s condition) to observe the extent of interictal epileptic discharges (IEDs). The epileptologists decide (or diagnose) SOZ channels in order to carry out epilepsy surgery via an epileptic focal resection. In this study, we explored a method of epileptic band and feature selection to design a system for identifying SOZ channels in HFCs (>80 Hz) of iEEGs. The first results highlighted the SVM and LightGBM classification performance to detect individual segments with different conditions (FbFM and proposed FBFM/Sb/FS). We obtained consistent performances with only 35 min of signals in the interictal phase. That made the proposed FbFM/Sb/FS using a LightGBM classifier particularly interesting for the identification of SOZ electrodes. Finally, we simulated the results to identify the possible electrodes related to SOZ from multi-channel iEEGs, which can help epileptologists to hypothesize the SOZ and non-SOZ channels. In our dataset, eight patients were “seizure-free” after surgery. Three patients (Pt5, Pt9, and Pt10) were “residual”, even though the SOZ labeled by epileptologists were removed after surgery. For residual patients (Pt5, Pt9, and Pt10), our computer-aided solution also suggested some channels, which were very close to the SOZ. This suggestion may assist epileptilogists to make a more confident decision about SOZ channels.

Moreover, we compared the proposed design with patient-independent design (PID). First, the results showed very poor performance for PID compared to the proposed PDD. Second, the AUC of PID for the patients (Pt4, Pt9, and Pt10) was less than 0.60. The possible reasons of the poor performance may be the very different locations of electrodes and the subject-specific nature of EEG signals. Compared to the proposed prior-based design, the major suitability of PID is that it requires no training data for new patients to detect SOZ channels. For identifying the SOZ, Varatharajah et al. [35] proposed an AI-based analytic framework with an SVM classifier, which utilizes multiple intracranial electrophysiological biomarkers such as HFO, IED, and phase-amplitude coupling (PAC) between low-frequency bands (0.1–30 Hz) and high-frequency bands (65–115 Hz). They used 82 patients to evaluate their system. They reported the performance of their framework with an average AUC of 0.79 for the cross validation of mixed data and an AUC of 0.73 for leave-one-patient-out cross-validation. For mixed data, they mixed the SOZ and NSOZ electrodes of all subjects to form a dataset. A limitation of their work was the fact that they only used partial ripple bands. In our patient-independent framework, we used high-frequency components such as ripple and fast ripple bands and an average AUC of 0.68. Although we have used a much smaller number of patients (only eleven patients) to evaluate the method, our result is very close to the work proposed by Varatharajah et al. [35]. In fact, our proposed study does not use biomarkers such as HFO, IED, and PAC. These biomarkers are considered very important in conventional clinical studies. However, from a machine learning point of view, statistical features, including information theoretic features, are very important to characterize epileptic signals. So, our statistical features are very good candidates of biomarkers (HFO, IED, and PAC). Moreover, there is still room for improvements of PID in terms of AUC. Future works could study more advanced signal processing methods by increasing the number of patients to improve the performance of the system. A more promising direction could be the use of domain transfer to adapt the different distributions [71].

For a direct comparison of other epilepsy-related studies to design an SOZ detection system, we have summarized the computer-aided solution under three different categories, including LFC- [12,13,14,35], HFC- [16,17], and HFO-related [23,24,25] research during the past decade (see Table 4). In epilepsy focal detection, we also referred to SOZ, Sharma et al. [12], who introduced a system for the identification of focal and non-focal iEEG signals using the Bern–Barcelona dataset, with an 87% accuracy. However, the Bern—Barcelona dataset consists of approximately 20 s iEEGs with a pair (two-channel iEEG) of focal and non-focal channels digitally band-pass filtered between 0.5 and 150 Hz, using a fourth-order Butterworth filter. The bi-variate focal and non-focal channels were recorded in the epileptic and non-epileptic zones of the brain. Similar problems with this dataset also led other authors to propose an improvement of the system performance using decomposition methods such as EMD and various classifiers, including Naïve Bayes (NBC), radial basis function (RBF), SVM, k-NN, non-nested generalized exemplars (NNge), and best first decision tree (BFDT) [13,14,20,72]. However, the EMD-based decomposition is only suitable for a single channel. When there is more than one channel in an EEG, the number of decomposed bands among channels is not consistent, putting a limit on the system in real applications. To address this problem, Itakura et al. proposed a bivariate EMD, improving classification accuracy of 1.73% compared to the EMD-based approach [14,20]. Indeed, both algorithms (EMD and BEMD) in their analysis had a lower mean accuracy compared to the original authors’ results. Compared to the above LFC-related studies, the novelty of the proposed design can be summarized as follows. First, we used the ripple and fast ripple frequency bands by utilizing the recent findings [26,27,28] rather than the lower frequency bands (0.5–150 Hz) used in the Bern–Barcelona dataset [73,74,75]. Second, a pair of focal and non-focal channels (balanced problem) were used in LFC-related studies. Due to a balanced problem, the previous studies used classification accuracy to evaluate the system performance. In a clinical situation, the number of SOZ and non-SOZ channels depends on patients, which creates an imbalanced problem. Therefore, to design the proposed computer-aided solution, we considered the multi-channel imbalance problems, which support the use of fully clinical applications.

In addition, the other LFC-related epilepsy studies used Bonn [78], Freiburg [79], and CHB-MIT datasets [75] and tried to diagnose whether an EEG segment has a seizure or non-seizure. In contrast, our methods allow us to detect or discriminate electrodes related to SOZ. This localization allows the surgeon to remove them for seizure freedom.

In HFO-related epilepsy studies, some research works [24,25,77] have proposed automated HFO detectors by considering the hypothesis that the rate of HFOs tends to be higher in SOZ. However, a major limitation of the HFO-related studies was that they not only used long-term EEG data to calculate the baseline, but also developed automated systems separately for individual ripples and fast ripples to find SOZ channels. Recently, our published studies used eight types of entropy features with the SVM method to design a SOZ detection system [16]. Tenfold cross-validation was used to evaluate the system in which each fold consisted of a total of nine segments. The average performance of the system showed a sensitivity of 52.7%, specificity of 90.75% for segment detection and an AUC of 0.86% for channel identification. Due to some limitations of that system, which we already addressed in the Introduction, we proposed a statistical feature-based computer-aided method to identify SOZ electrodes. The proposed system achieved an average sensitivity of 85.73% and a higher specificity of 97.50% for segment detection and an average AUC of 0.99 for channel identification. Note that the proposed method used a combination of 60 segments in the test phase to identify the channels. The average computational time with 12 feature extraction methods to identify a segment was 0.75 s. Akter et al. reported that the average computational time of a system with eight entropies and ten subbands was 56.51 s to test a single segment for 60 channels [16]. However, assuming that the testing data have 60 segments, the system’s computation cost would be close to 60 min to identify SOZ and non-SOZ channels. The proposed method could facilitate this procedure (in only 0.2 min) without compromising performance, improving the system’s usability.

## 5. Conclusions

This study developed an effective epileptic channel identification method using HFCs in interictal iEEG data. The twelve types of statistical feature extraction methods with filter bank analysis were used. The joint contribution of bands and features was selected based on MI-based approach with a grid search to improve the performance of the proposed method for identifying possible channels with epileptic events in multi-channel iEEGs. The proposed methods were evaluated by eleven medically intractable epilepsy patients with focal cortical dysplasia (FCD). The experimental results indicated that the proposed statistical feature-based computer-aided method can efficiently identify SOZ electrodes in the HFCs of iEEGs. 

## Figures and Tables

**Figure 1 entropy-22-01415-f001:**
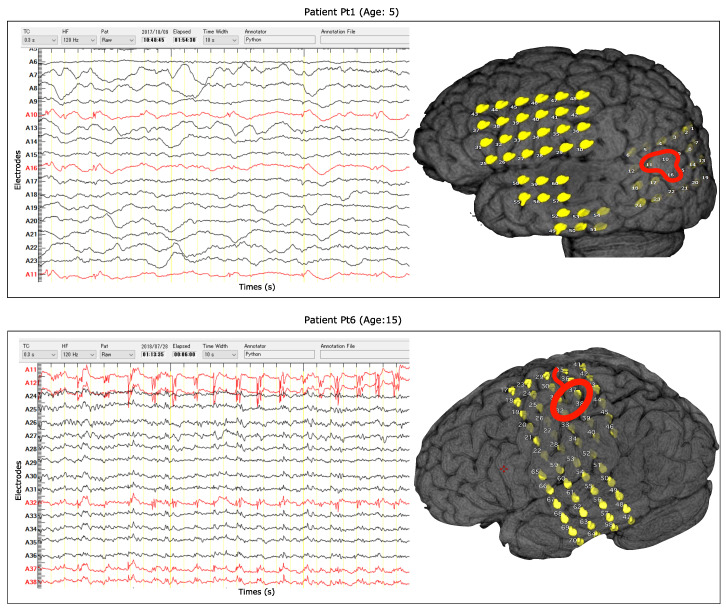
The intracranial electroencephalogram (iEEG) signals and their magnetic resonance imaging (MRI) images with 17 electrodes for patients Pt1 and Pt6. The red circles indicate the SOZ electrodes labeled by clinical experts.

**Figure 2 entropy-22-01415-f002:**
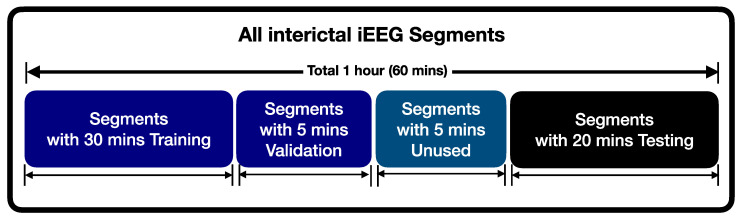
The allocation of iEEG data used for training and testing in the proposed method.

**Figure 3 entropy-22-01415-f003:**
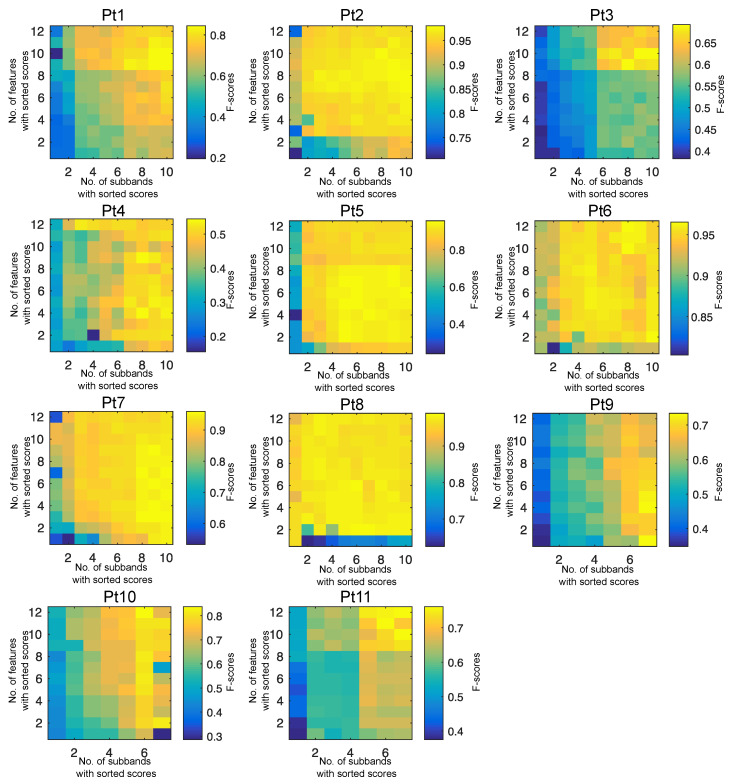
Grid parameter search for optimizing features and subbands. The F-score was obtained as a joint combination of features (y-axis) and subbands (x-axis) with maximum MI scores derived from Equations (Equation 22) and (Equation 24).

**Figure 4 entropy-22-01415-f004:**
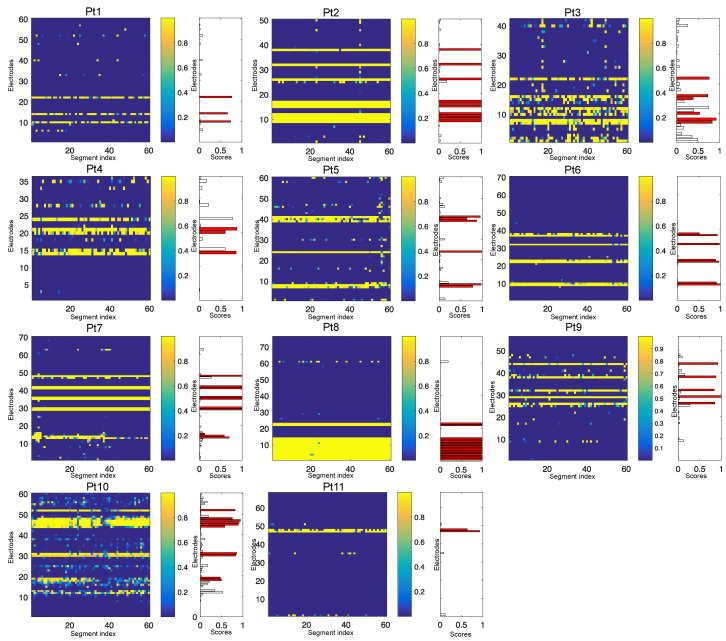
The simulated results using the FbFM/Sb/FS method to identify the seizure onset zone (SOZ) channels. For each patient, the color map (left) and average value of scores with channels achieved from each segment using the proposed method (right) are presented. The red color bars indicate the SOZ channels labeled by epileptologists.

**Figure 5 entropy-22-01415-f005:**
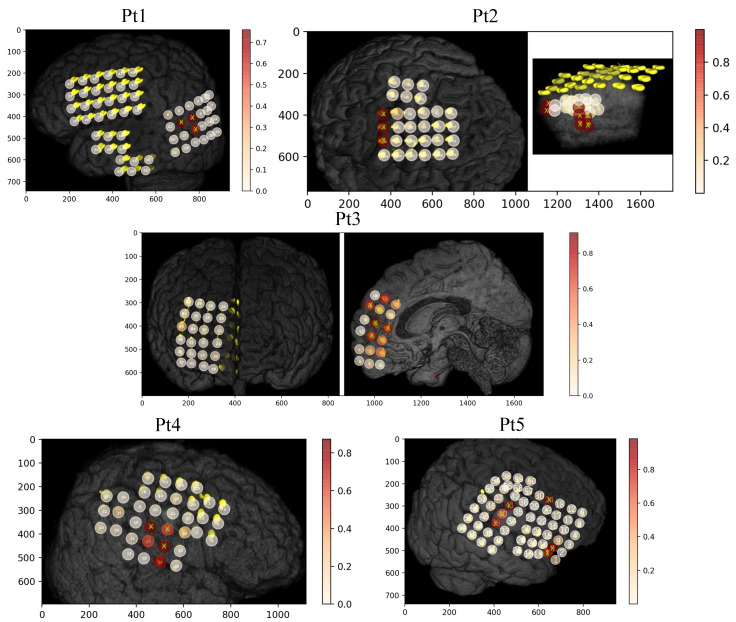
MRI scan images with electrode positions for 5 patients (Pt1 to Pt5). The electrodes of SOZ channels are represented by “X”, labeled using clinical expertise, and the circles with color indicate the magnitude of each channel score achieved by the proposed method.

**Figure 6 entropy-22-01415-f006:**
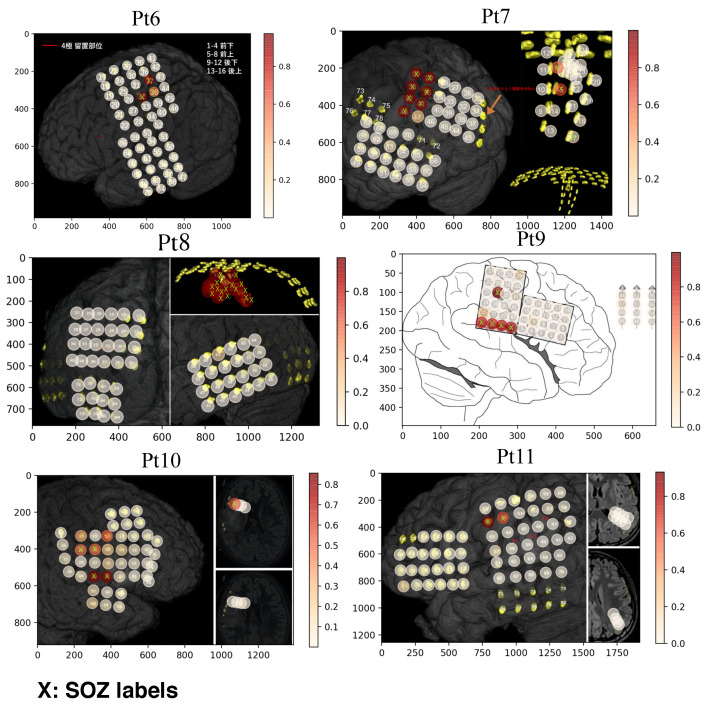
MRI scan images with electrode positions for 6 patients (Pt6 to Pt11). The electrodes of SOZ channels are represented by “X”, labeled using clinical expertise, and the circles with color indicate the magnitude of each channel score achieved by the proposed method.

**Figure 7 entropy-22-01415-f007:**
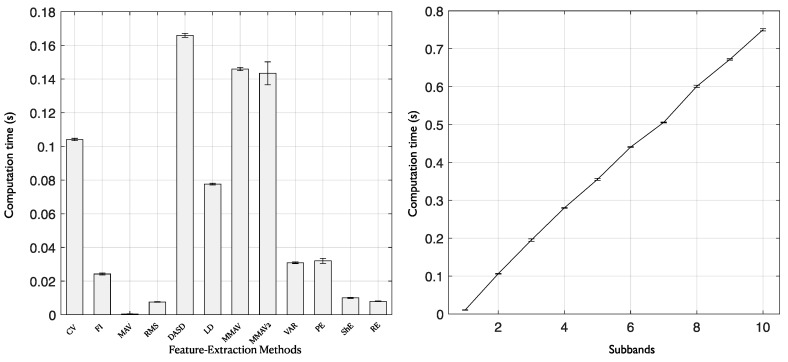
Average computational time with each feature extraction method for 10 subbands (**Left**). Average computational time with number of subbands for feature extraction methods (**Right**).

**Table 1 entropy-22-01415-t001:** The summary of interictal iEEG data for individual patients with focal cortical dysplasia (FCD) [39]. The males and females are indicated as M and F.

Patients ID	Age and Sex	Lesion Site	Location	Pathology	Sampling Frequency	No. of Electrodes	No. of SOZ Electrodes	Follow up	Engel
**Pt1**	5/F	Lt dorsal superior temporal gyrus	Cortical surface	Type 2B	2 KHz	60	3	3 years	IA
**Pt2**	39/F	Lt dorsal superior frontal gyrus	Bottom of sulcus	Type 2B	2 KHz	50	10	3 years	IA
**Pt3**	5/M	Lt cingulate gyrus	Bottom of sulcus	Type 2B	2 KHz	42	6	3.5 years	IA
**Pt4**	6/M	Rt dorsal middle frontal gyrus	Cortical surface	Type 2B	2 KHz	36	3	3.5 years	IA
**Pt5**	20/M	Rt middle frontal gyrus	Cortical surface	Type 2A	2 KHz	60	6	4.5 years	IIIA
**Pt6**	15/M	Lt superior parietal lobule	Cortical surface	Type 2B	2 KHz	70	7	5 years	IA
**Pt7**	32/M	Lt superior parietal lobule	Bottom of sulcus	Type 2B	2 KHz	70	10	5 years	IA
**Pt8**	25/M	Lt angular gyrus	Bottom of sulcus	Type 2A	2 KHz	76	16	5 years	IA
**Pt9**	38/F	Rt supramarginal gyrus	Surface and vertical cortex	Type 2B	1 KHz	56	5	5.5 years	IIA
**Pt10**	14/F	Rt inferior frontal gyrus	Cortical surface	Type 2B	1 KHz	60	10	5.5 years	IC
**Pt11**	13/M	Lt angular gyrus	Surface and vertical cortex	Type 2B	1 KHz	68	2	5 years	IA

**Table 2 entropy-22-01415-t002:** Experimental results for individual segment detection with eleven patients using Filter bank Feature Extraction Method (FbFM) and proposed FbFM/Sb/FS. In both cases, we used support vector machine (SVM) and LightGBM to detect individual segments. The three evaluation metrics were included: Sen (sensitivity), Spe (specificity), and $F_{1}$ score (*F*-score). The numbers in bold represent the maximum performance.

Methods	Evaluation Matrics	Patient ID											
		Pt1	Pt2	Pt3	Pt4	Pt5	Pt6	Pt7	Pt8	Pt9	Pt10	Pt11	Avg
	**Sen**	62.77	84.00	58.61	34.44	67.50	90.48	75.50	92.60	70.66	47.66	86.66	70.08
**FbFM**	**Spe**	99.32	94.87	87.92	**93.78**	97.62	94.47	97.88	99.53	91.53	91.80	97.14	95.08
**with SVM**	**$F_{1}$ score**	0.71	0.82	0.50	0.33	0.71	0.75	0.80	0.95	0.55	0.51	0.62	0.67
	**Sen**	73.33	99.39	**81.94**	77.78	69.44	**90.95**	81.83	97.81	69.33	53.16	**89.16**	80.42
**FbFM/Sb/FS**	**Spe**	99.39	96.42	87.59	93.08	97.78	96.24	97.41	99.72	95.98	95.43	**98.36**	96.13
**with SVM**	**$F_{1}$ score**	0.79	0.93	**0.63**	0.61	0.73	0.80	0.82	0.98	0.66	0.60	0.73	0.76
	**Sen**	78.33	99.33	67.50	76.67	86.11	90.00	83.33	92.70	88.66	74.5	83.33	83.68
**FbFM**	**Spe**	98.62	97.16	85.60	91.11	97.59	**99.80**	97.16	99.52	93.36	95.8	99.06	95.89
**with LGBM**	**$F_{1}$ score**	0.76	0.94	0.53	0.55	0.82	0.94	0.83	0.95	0.69	0.76	0.77	0.78
	**Sen**	**82.78**	**99.83**	67.77	**81.11**	**90.83**	90.24	**83.83**	**99.69**	**89.33**	**76.83**	80.33	**85.73**
**FbFM/Sb/FS**	**Spe**	**99.44**	**98.92**	**90.09**	93.48	**98.20**	99.77	**98.78**	**99.69**	**98.20**	**96.33**	99.67	**97.50**
**with LGBM**	**$F_{1}$ score**	**0.86**	**0.98**	0.60	**0.64**	**0.88**	**0.95**	**0.88**	**0.99**	**0.86**	**0.79**	**0.84**	**0.84**

**Table 3 entropy-22-01415-t003:** The results of AUC to detect SOZ channels for both patient-dependent and -independent designs. The numbers in bold represent the maximum performance.

Patient ID	AUC	
	Dependent	Independent
**Pt1**	1.00	0.77
**Pt2**	1.00	0.72
**Pt3**	0.97	0.64
**Pt4**	0.98	0.55
**Pt5**	1.00	0.66
**Pt6**	1.00	0.63
**Pt7**	0.99	0.77
**Pt8**	1.00	0.65
**Pt9**	1.00	0.58
**Pt10**	0.99	0.55
**Pt11**	1.00	0.98
**Mean**	**0.99**	0.68

**Table 4 entropy-22-01415-t004:** Comparison with proposed the system with other existing systems. The sentences in bold represent the proposed method.

Studies	Ref	Dataset	Methods	Bands	Goal	Performance
**LFC**	Sharma et al. [12]	Bern Barcelona	-EMD 6 entropy -based features -LS-SVM	-Lower bands (0.5–150 Hz)	Epileptic focus detection	ACC: 88%
	Arunkumar N. et al. [13]	Bern Barcelona	3 entropy -based features -NBC, SVM, k-NN, RBF NNge, BFDT	-Lower bands (0.5–150 Hz)	Epileptic focus detection	ACC: 98.0%; Sen: 100%; SPE: 96.0%
	Itakura et al. [14]	Bern Barcelona	-BEMD 6 entropy -based features -LS SVM	-Lower bands (0.5–150 Hz)	Epileptic focus detection	ACC: 86.0%
	Varatharajah et al. [35]	Mayo Clinic	-PAC, HFOs, IEDs -SVM	-Lower bands (0.1–30 Hz) -Partial ripple bands (65–115 Hz)	SOZ detection	For cross-validation of mixed data AUC: 0.79 and For leave one patient out cross-validation AUC: 0.73
	Yang Y. et al. [76]	Bern Barcelona	-FAWT -2 entropy -based features -GRNN, SVM, LS-SVM, k-NN fKNN	-Lower bands (0.5–150 Hz)	Epileptic focus detection	ACC: 94.80%
**HFO**	Jrad et al. [24]	Rennes University Hospital	-Gabor transformation -Energy-based feature -SVM	-Ripple (120–250 Hz) -Fast ripple bands (250–600 Hz)	HFOs detection	Ripple-HFOs (Sen: 81.1% and FDR: 30.2%) and Fast ripple-HFOs (Sen: 74.6% and FDR: 6.3%)
	Zuo et al. [25]	Xuanwu Hospital	-Deep CNN -Time–frequency map	-Ripple (80–250 Hz) -Fast ripple bands (250–500 Hz)	HFOs detection	Ripple-HFOs (Sen: 77.0% and SPE: 72.3%) and Fast ripple-HFOs (Sen: 83.2% and SPE: 79.3%)
	Lai et al. [77]	West China Hospital	-CNN method	-Ripple (80–250 Hz) -Fast ripple bands (250–500 Hz)	HFOs detection	Ripple-HFOs (Sen: 82.2% and FDR: 12.6%) and Fast ripple-HFOs (Sen: 93.4% and FDR: 8.0%)
**HFC**	Akter et al. [16]	Juntendo Hospital	-Multiband -Eight types of entropy -Feature selection -SVM	-Ripple and fast ripple bands (100–600 Hz) -Eight patients −2 kHz	SOZ detection	Segment Detection (Sen: 52.7%; Spe: 90.75%) and SOZ detection (AUC = 0.86) Com. time: 56.51 s
	**Proposed**	**Juntendo** **Hospital**	-**Multiband** -**12 types of features** -**Joint bands and** **feature selection** -**LightGBM**	-**Ripple and** **fast ripple** **bands (100–600 Hz)** **and** **bands (100–450 Hz)** -**Eleven patients** −**2 kHz and 1 kHz**	**SOZ** **detection**	**Segment detection** **(Sen: 85.73%; Spe: 97.50%)** **and** **SOZ detection (AUC = 0.99)** **Com. time: 0.8 s**

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
