# Peer review of "Statistical Features in High-Frequency Bands of Interictal iEEG Work Efficiently in Identifying the Seizure Onset Zone in Patients with Focal Epilepsy"

_entropy, 2020, doi:10.3390/e22121415_

Round 1
Reviewer 1 Report
The research content of the manuscript is very meaningful, but the manuscript needs to be thoroughly revised, especially the language part.
- The tenses of sentences in the manuscript need to be unified, such as the following sentence. “To achieve more accurate detection, several studies proposed subband decompositions with multiple filters that have different passbands [17,40,41].”
- There are some obvious grammatical errors in the manuscript, such as the following sentence. “Our recent study in epilepsy focus detection first proposed a filter-bank approach to divide the HFCs (ripple and fast ripple) into multiple subbands [16,17].”
- The sentences in the manuscript need to conform to the writing habits of English papers, otherwise readers will misunderstand the original meaning, such as the following sentence. “For the patients with 1 kHz sample frequency, the highest cut-off frequency should keep half of the sampling rate to reconstruct signals by reducing the aliasing effect [42].”
- Many existing studies have been cited in the manuscript, but the most advanced study should also be referred to. https://doi.org/10.1016/j.bspc.2020.102279
- Punctuation marks should be used in accordance with English writing requirements, such as the following sentence. “For x, the coefficient of variation can be defined as [9,10]:”
- Two formulas in (2) cannot be marked with only one serial number.
Reviewer 2 Report
The present research is really original and interesting in the field of EEG analysis in epilepsy.
Authors have correctly described the complex algorithms underlying these analyzes.
Method devised and proposed by the authors for the evaluation of the epileptogenic focus is truly original.
The methods are described correctly, as well as the results are correctly interpreted.
The discussion and the Introduction clearly express the theoretical assumption and what is reported in the literature.
References are updated and well focused on research topic.
English is good and easy to read.
No changes are required
This manuscript is a resubmission of an earlier submission. The following is a list of the peer review reports and author responses from that submission.
Round 1
Reviewer 1 Report
The authors have analyzed the statistical features in high-frequency bands of interictal intracranial EEG epileptiform discharges in identifying the seizure onset zone in eleven patients who underwent epilepsy presurgical evaluation at their center.
The main issue with this research article is the lack of clinical novelty in this field which has been enormously studied already in this field (for references see for example the study of Tamilia et al. "Surgical resection of ripple onset predicts outcome in pediatric epilepsy." Annals of neurology 84.3 (2018): 331-346.). Another important flaw is represented by the results which are not replicable for clinical use.
Furthermore, there are major methodological and design issues that need to be addressed. The writing in English language needs a deep revision too (some sentences are difficult to understand). Below my comments:
1) Title: The term “epileptic seizure onset zone” is inappropriate (does a non-epileptic seizure onset zone do exist?). Just keep “seizure onset zone”. Also, the title is very long and hard to understand.
2) Introduction: The introduction is very long and does not get to the point until the very last paragraph. Only 3 not recent studies from the enormous literature concerning HFO detection and clinical value for epilepsy surgery are reported. A thorough revision and summary of this section is warranted. Finally, all patients analyzed had focal cortical dysplasia (FCD). If this is intentional it should be mentioned in the study aims and a detailed explanation needs to be provided (why only patients with FCD?).
3) Table 1: What is the meaning of the column “Location”? What do the authors mean with “Bottom”? Does this mean deep cortical source? Please provide the exact cortical location of the FCD at the sublobar level (i.e. deep frontal, mesial temporal, etc.). You should specify the type of inctracranial EEG electrodes employed for each patient (stereo-EEG? Subdural electrodes (grids, strips)?, ECOG? A combination of both?). Outcome should be reported using the Engel epilepsy surgery oucome classification (the term “Residual” is not acceptable, what does it mean? Less than 50% reduction? Same frequency as before? Just a few seizures?). Also, how long is the follow-up? A follow-up inferior to 1 year is not acceptable to determine the clinical outcome of these patients.
4) Data and Methods: The authors did not mention whether they analyzed awake or sleep EEG signals. Also on page 9 line 204 they say that they analyzed “interictal iEEG data”, but there is no mention of interictal epileptiform discharges (IED) identification (who identified for example spikes? What type of IED did they find in their patient? How many IEDs per patients?). These elements are of crucial importance since the presence of IED is of crucial importance in the interpretation and analysis of HFO.
5) Results: The most concerning aspect of this study is the statistical analysis. By using the SVM and Light GBM to determine the optimal threshold that differentiates SOZ channels this approach qualifies as a feature optimization method. Whenever you do feature optimization, you must do cross validation. Otherwise the results are not generalisable. In other words: Your results are valid only for the presented sample but are not replicable. This is a severe issue and the major concern I have for this article. In order to be able to cross validate, you can do the crossvalidation with k-fold cross validation or leave-one-out cross validation. k-fold is in general favourable but it is probably a bit problematic in this small sample. Another important concern is that not all patients had good surgical outcome so the SOZ channels that they considered for the machine-learning classifier are most likely not the true SOZ channels in 3 patients (if they keep having seizure, this means that the EZ was not completely resected). These patients should not be included in the model for these obvious reasons.
Author Response
I would like to express our gratitude to the reviewer for constructive comments and suggestions.
Our team has considered updating the manuscript with the comments that need to change for improving the quality of the manuscript highlighted with blue color text.
The followings are point-to-point answers to the comments.
The authors have analyzed the statistical features in high-frequency bands of interictal intracranial EEG epileptiform discharges in identifying the seizure onset zone in eleven patients who underwent epilepsy presurgical evaluation at their center.
The main issue with this research article is the lack of clinical novelty in this field which has been enormously studied already in this field (for references see for example the study of Tamilia et al. "Surgical resection of ripple onset predicts outcome in pediatric epilepsy." Annals of neurology 84.3 (2018): 331-346.). Another important flaw is represented by the results which are not replicable for clinical use.
Response: First of all, we would like to emphasize that our study aims to develop data-driven methodologies to identify the SOZ channels, not to find novel epileptic biomarkers for surgical resection. The final goal of this study is to establish a computer-aided system for clinicians to identify the SOZ.
Please note that for designing a computer-aided system, recent studies used either biomarkers, including high-frequency oscillations (HFOs) [1,2], phase-amplitude Coupling (PAC) [3,4], Interictal Epileptiform Discharges (IEDs) [5,6,7] or use of feature-extraction methods [8,9,10]. Several studies used the feature-extraction methods to design AI systems [8,9,10], which provides evidence to support the use of feature-extraction methods in iEEG signals. So, we are the first one to introduce the statistical features to design such AI-systems for localizing the possible SOZ channels.
The reviewer refers to the manuscript title "Surgical resection of ripple onset prediction outcome in pediatric epilepsy," published in Annals of Neurology, 2018 (Tamilia et al., 2018). The manuscript’s main objective was to assess the spatiotemporal propagation of interictal ripples (ripple HFOs) on iEEG and compared it with the propagation of spikes (IEDs) to evaluate their clinical value as epilepsy biomarkers. Therefore, their study mainly focuses on epilepsy biomarkers by statistical analysis of distinctive attributes of ripple HFOs and interictal spikes (IEDs) in iEEG. Finally, they observed that onset-ripples are new promising biomarkers of the epileptogenic zone (EZ) compared to spread-ripples and onset-spikes.
We understand there are many works on epileptic biomarkers like the suggested paper Tamilia et al. (2018). However, our paper has a different purpose. We aim to develop a machine learning-based diagnostic aid for identifying the SOZ without using biomarkers like HFOs or IEDs. More specifically, our study focuses on designing a computer-aided solution using statistical features so that the epileptologists can make a quick hypothesis about SOZ channels by observing the score of the multiple channels rather than manually observing long-term multi-channel iEEG signals. As discussed in the Introduction, many related works based on AI aim to develop automated methods for finding such biomarkers. Besides, many techniques from the engineering side to find epileptic focus are based on only public datasets (most of them are the so-called Bern-Barcelona dataset) that lack meta-data.
Again, the current study is mainly related to technological-based research and was evaluated based on the clinical labeled-data similar to previous AI-based studies. Our contributions could be three-fold: a) developing statistical feature values that fit simple machine learning techniques, b) using in-hospital iEEG data with medical backgrounds, and c) showing a novel representation with channel scores, assisting the data analysis for epileptologists.
References:
- Crépon, B., Navarro, V., Hasboun, D., Clemenceau, S., Martinerie, J., Baulac, M., Adam, C., Le Van Quyen, M., 2010. Mapping interictal oscillations greater than 200 Hz recorded with intracranial macroelectrodes in human epilepsy. Brain 133, 33– 45.
- Jrad, N.; Kachenoura, A.; Merlet, I.; Bartolomei, F.; Nica, A.; Biraben, A.; Wendling, F. Automatic Detection and Classification of High-Frequency Oscillations in Depth-EEG Signals. IEEE Transactions on Biomedical Engineering 2017, 64, 2230–2240.
- M. Amiri, B. Frauscher, and J. Gotman. Phase-amplitude coupling is elevated in deep sleep and in the onset zone of focal epileptic seizures. Frontiers in Human Neuroscience 10(387), 2016.
- Bahareh Elahian, Mohammed Yeasin, Basanagoud Mudigoudar, James W. Wheless, and Abbas Babajani-Feremi. Identifying seizure onset zone from electrocorticographic recordings: A machine learning approach based on phase locking value. Seizure, 51:35–42, 2017.
- J. Gotman and P. Gloor, ‘‘Automatic recognition and quantification of interictal epileptic activity in the human scalp EEG,’’ Electroencephalogr. Clin. Neurophysiol., vol. 41, no. 5, pp. 513–529, Nov. 1976.
- J. Puspita, G Soemarno, A. Jaya, and E Soewono, “Interictal epileptiform discharges (IEDs) classification in eeg data of epilepsy patients”, in Journal of Physics: Conference Series, vol. 943, 2017, p. 012 030.
- Varatharajah, Y.; Berry, B.; Cimbalnik, J.; Kremen, V.; Gompel, J.V.; Stead, M.; Brinkmann, B.; Iyer, R.; Worrell, G. Integrating artificial intelligence with real-time intracranial EEG monitoring to automate interictal identification of seizure onset zones in focal epilepsy. Journal of Neural Engineering 2018, 15, 046035.
- Sharma, R.; Pachori, R.B.; Acharya, U.R. Application of Entropy Measures on Intrinsic Mode Functions for the Automated Identification of Focal Electroencephalogram Signals. Entropy 2015, 2, 669–691.
- Arunkumar, N.; Ramkumar, K.; Venkatraman, V.; Abdulhay, E.; Fernandes, S.L.; Kadry, S.; Segal, S. Classification of focal and non focal EEG using entropies. Pattern Recognition Letters 2017, 94, 112–117.
- Akter, M.S.; Islam, M.R.; Iimura, Y.; Sugano, H.; Fukumori, K.; Wang, D.; Tanaka, T.; Cichocki, A. Multiband Entropy–based Feature–extraction Method for Automatic Identification of Epileptic Focus based on High-frequency Components in Interictal iEEG. Scientific Reports 2020, 10, 1–17.
Furthermore, there are major methodological and design issues that need to be addressed. The writing in English language needs a deep revision too (some sentences are difficult to understand).
Response: Let us explain our fundamental concept of this paper. In conventional medical systems, to localize the possible SOZ channels, one approach is that the epileptologists need to observe long-term multi-channel iEEG data in time-series forecasting. This process is challenging and time-consuming for the epileptologists.
To address this problem, we represented the multi-channel long-term iEEGs into channel indexes so that epileptologists can hypothesize the possible SOZ channels in a much easier way. To this end, our methodology and engineering ML design included the following step:
(1) Use of multi-channels iEEG data with high-frequency bands, 2) Apply signal processing methods to the high-frequency multi-channels iEEG data, 3) Feature-extraction from the processed high-frequency multi-channels iEEG data, 4) Solution of imbalanced problem, 5) Scoring the segments in the way of time-series forecasting based on ML-methods, and finally 6) Providing the scoring to the multi-channels iEEGs estimated from the segments in the way of time-series forecasting.
The development of an AI solution based on the above six steps is our design that has been expected by epileptologists for localizing the possible SOZ channels.
Regarding English, before submitting the manuscript to the entropy journal, we proofread our manuscript by an English professional. If the Reviewer could kindly give us direction (we do not find any direction in the reviewer comments below), which parts or sentences are difficult to understand, we consider updating them to our manuscript.
Comment 1. Title: The term “epileptic seizure onset zone” is inappropriate (does a non-epileptic seizure onset zone do exist?). Just keep “seizure onset zone”. Also, the title is very long and hard to understand.
Response: Thank you so much for the comment. We agree with the Reviewer’s comment to remove "epileptic" before "seizure onset zone."
We have also updated the title in our manuscript following:
“Statistical features in high-frequency bands of interictal iEEG work efficiently in identifying the seizure onset zone in patients with focal epilepsy”
Comment 2. Introduction: The introduction is very long and does not get to the point until the very last paragraph. Only 3 not recent studies from the enormous literature concerning HFO detection and clinical value for epilepsy surgery are reported. A thorough revision and summary of this section is warranted. Finally, all patients analyzed had focal cortical dysplasia (FCD). If this is intentional it should be mentioned in the study aims and a detailed explanation needs to be provided (why only patients with FCD?).
Response: Our research aim was to develop a machine-learning methodology that can assist the epileptologists. We focus on our related studies in the introduction’s flow, including their procedures, design issues, and real-world applications’ limitations. In our introduction parts, we included brief reviews of three biomarkers related studies. Thus, several studies proposed to use these biomarkers as computer-aided solutions to detect the HFOs. In our introduction parts, we included 11 HFO-based computer-aided solution related studies. Since our research did not focus on finding new biomarkers or developing the computer-aided solution using these biomarkers, the increase of biomarkers-related studies or HFO-automated studies can misguide the main aim of our research to the reader. However, the HFOs-related studies provided us evidence to use high-frequency bands in interictal iEEGs to design our computer-aided solutions. We offered this message (evidence to use high-frequency bands in iEEG) to the reader through paragraph 4 of our introduction section. So far, our research team believes that the flow of introduction to the manuscript is convenient to the reader.
Why only patients with FCD?
In FCD, IED sites (spike) appearance and seizure initiation are centered on imaging sites. A high seizure suppression rate has been reported in FCD type 2, particularly by excising the abnormal imaging sites. We chose FCD because of its potential to homogenize or improve the accuracy of the teacher data. However, our study aimed to establish a machine learning methodology to identify the seizure onset zone. For evaluation, the type of epilepsy is preferred to be identical. We chose FCD because it is relatively easy to identify the seizure onset zone with MR imaging only. This means that the reliability of labels in machine learning is high, which was suitable for evaluating machine learning algorithms.
Comment 3. Table 1: What is the meaning of the column “Location”? What do the authors mean with “Bottom”? Does this mean deep cortical source? Please provide the exact cortical location of the FCD at the sublobar level (i.e. deep frontal, mesial temporal, etc.). You should specify the type of inctracranial EEG electrodes employed for each patient (stereo-EEG? Subdural electrodes (grids, strips)?, ECOG? A combination of both?). Outcome should be reported using the Engel epilepsy surgery oucome classification (the term “Residual” is not acceptable, what does it mean? Less than 50% reduction? Same frequency as before? Just a few seizures?). Also, how long is the follow-up? A follow-up inferior to 1 year is not acceptable to determine the clinical outcome of these patients.
Response: Thank you for these comments to improve the quality of our manuscripts. We have added information in Table 1 with the “lesion site,” “location,” “follow up,” and “Engel” to our manuscript to respond to the query. Please check Table 1 in our manuscript.
We add the following sentences to the dataset section in our manuscript to respond to the type of interictal iEEG as following:
“To record data, the epilepsy surgeon implanted platinum subdural grids (UNIQUE MEDICAL Co, Tokyo, Japan) with 4-mm diameter and 10-mm distance for cortical surface and platinum strip electrodes (UNIQUE MEDICAL Co, Tokyo, Japan) with 3-mm diameter and 5-mm distance for vertical and bottom of cortex.”
We have also included the following information to the dataset section in the manuscript following:
“The seizure-free outcomes were treated as class IA in Engel’s classification. The mean follow up period was 4.9 ±1.0 years. Seizure outcomes were evaluated using Engel’s classification at the last visit to the outpatient center.”
Comment 4. Data and Methods: The authors did not mention whether they analyzed awake or sleep EEG signals. Also on page 9 line 204 they say that they analyzed “interictal iEEG data”, but there is no mention of interictal epileptiform discharges (IED) identification (who identified for example spikes? What type of IED did they find in their patient? How many IEDs per patients?). These elements are of crucial importance since the presence of IED is of crucial importance in the interpretation and analysis of HFO.
Response: Thank you for the comments. We selected the sleep stage iEEG without motion artifact for analysis. Epileptologists picked 60-min iEEG during sleep without artifact. In the case of our study, we did not observe the type of IEDs. However, the selected iEEG may contain various IEDs, including spike, wave complex, polyspike, and slow-wave component. Since our study does not focus on using biomarkers like IEDs and HFOs to design the computer-aided solution, counting IEDs or analysis of HFOs was not crucial in our research. And this is a significant advantage to use the feature-extraction methods compared to biomarkers like IEDs or HFOs related systems.
We have included the following sentence in the dataset section of our manuscript as:
“Epileptologists selected the sleep stage iEEG without motion artifact for analysis.”
Also, we have added some information in the dataset section to our manuscript to avoid confusion as follows:
“They were assigned the positive labels to the SOZ electrodes for each patient, and a negative label was given to the rest of the electrodes. Therefore, data obtained from SOZ and non-SOZ electrodes were used to design the proposed computer-aided solution.”
Comment 5. Results: The most concerning aspect of this study is the statistical analysis. By using the SVM and Light GBM to determine the optimal threshold that differentiates SOZ channels this approach qualifies as a feature optimization method. Whenever you do feature optimization, you must do cross validation. Otherwise the results are not generalisable. In other words: Your results are valid only for the presented sample but are not replicable. This is a severe issue and the major concern I have for this article. In order to be able to cross validate, you can do the crossvalidation with k-fold cross validation or leave-one-out cross validation. k-fold is in general favourable but it is probably a bit problematic in this small sample. Another important concern is that not all patients had good surgical outcome so the SOZ channels that they considered for the machine-learning classifier are most likely not the true SOZ channels in 3 patients (if they keep having seizure, this means that the EZ was not completely resected). These patients should not be included in the model for these obvious reasons.
Response: Sorry for the confusion. The SVM and Light GBM are machine-learning techniques used in our study to estimate each segment’s score. We did not use the SVM and LightGBM to determine the optimal threshold or feature optimization method. However, epileptologists need to observe long-term iEEG time series (3 to 7 days iEEG time series depending on the patients’ conditions). So, the model needs to identify the SOZ in the way of time series forecasting. Our research target was to provide a graphical illustration (intuition) with the index in time order to the medical experts. To this end, each iEEG segment was indicated as scores with the index in time order (please see figure 4). Finally, we sum all segment scores over time to give the weight to each channel. In recent ML research for time-series forecasting, k-fold, Leave-One-Out Cross-Validation (LOO) CV is not recommended [11,12,13]. It is standard to use the time-series CV.
The AI system gives some suggestions to the clinical experts about the disease for treatment. Based on the AI suggestion and different medical tests, the medical experts try to make confident medical decisions for treatments. For example, patient Pt10 was residual (that means the patient was not seizure-free after surgery). One possible reason is that clinical experts need to extend the SOZ’s boundary by observing again through post medical tests. But Interesting outcome with the proposed AI-system for patient Pt10 (as an example without seizure-free patient) was that our proposed AI system could provide some suggestions to extend the SOZ area. So clinical experts may carefully consider the AI suggestion with other medical tests to reestimate the SOZ.
We have included the following sentences in the division of the iEEG Time Series for Training and Testing section to our manuscripts for avoiding confusion:
"To evaluate the developed patient-dependent computer-aided solution, the division of the time series iEEG is the critical step. By considering the time series’s nature, we used the time-series cross-validation techniques [11,12,13], one of the system’s appropriate solutions to evaluate the model. We divided the iEEG data into training, validation, and testing in the way of time-series forecasting as illustrated in Fig. 2."
Reference:
11. C. Bergmeir and J. M. Benítez. On the use of cross-validation for time series predictor evaluation. Inf. Sci., 191:192–213, May 2012. ISSN 0020–0255.
12. L. J. Tashman. Out-of-sample tests of forecasting accuracy: an analysis and review. International Journal of Forecasting, 16(4):437–450, 2000.
13. S. Varma and R. Simon. Bias in error estimation when using cross-validation for model selection. BMC Bioinformatics, 7(1):91, Feb 2006.

Reviewer 2 Report
The manuscript shows how statistical features of high frequency iEEG bands can be used to provide computationally efficient solution to detecting SOZ.
The paper is well structured, easy to follow and provides a good overview of the state-of-the-art.
Particularly, I appreciate that the authors compared leave-one-out approach to a personally calibrated setup. Unfortunately, the first approach is often problematic in biosignals research and often avoided to push the presented results to better numbers. Here, the authors choose to report the results and discuss the limitations.
The introduction of statistics put on top of the frequency-based feature extraction is a good approach to increase the computational speed and in particular entropies tend to show good performance on EEG data.
Overall, I find a paper a very informative and relevant read.
Author Response
I would like to express our gratitude to the reviewer for constructive comments and suggestions.
Our team has considered updating the manuscript with the comments that need to change for improving the quality of the manuscript highlighted with blue color text.
The followings are point-to-point answers to the comments.
Comment 1: The manuscript shows how statistical features of high frequency iEEG bands can be used to provide computationally efficient solution to detecting SOZ.
The paper is well structured, easy to follow and provides a good overview of the state-of-the-art.
Particularly, I appreciate that the authors compared leave-one-out approach to a personally calibrated setup. Unfortunately, the first approach is often problematic in biosignals research and often avoided to push the presented results to better numbers. Here, the authors choose to report the results and discuss the limitations.
The introduction of statistics put on top of the frequency-based feature extraction is a good approach to increase the computational speed and in particular entropies tend to show good performance on EEG data.
Overall, I find a paper a very informative and relevant read.
Response: Thank you for your enthusiasm and positive comments about our manuscript. To evaluate the computer-aided solution for patient-independent design, we used the leave-patient-one-out (LPOO) CV. For patient-dependent design, we used the time-series CV. In our patient-dependent computer-aided solution, we have represented the long-term multi-channel time-series by the scores of channel indexes (please see Figure 4) so that epileptologists can only observe the scores of multi-channels to hypothesize the possible SOZ channels rather than observing long-term iEEG time series. And the scores of channel indexes should be estimated from each segment with the index in time order similar to the way of time-series forecasting.
To avoid confusion, we have included the following sentences with brief reviews in the division of the iEEG Time Series for Training and Testing section to our manuscripts:
“To evaluate the developed patient-dependent computer-aided solution, the division of the time series iEEG is the critical step. By considering the time series’s nature, we used the time-series cross-validation techniques [11,12,13], one of the system’s appropriate solutions to evaluate the model. We divided the iEEG data into training, validation, and testing in the way of time-series forecasting as illustrated in Fig. 2.”
Reference:
- C. Bergmeir and J. M. Benítez. On the use of cross-validation for time series predictor evaluation. Inf. Sci., 191:192–213, May 2012. ISSN 0020–0255.
- L. J. Tashman. Out-of-sample tests of forecasting accuracy: an analysis and review. International Journal of Forecasting, 16(4):437–450, 2000.
- S. Varma and R. Simon. Bias in error estimation when using cross-validation for model selection. BMC Bioinformatics, 7(1):91, Feb 2006.

Round 2
Reviewer 1 Report
Authors' changes did not substantially change the quality of the manuscript and my previous concerns.